# The Effect of Prebiotics, Alone or as Part of Synbiotics, on Cardiometabolic Parameters in Women with Polycystic Ovary Syndrome: A Systematic Review and Meta-Analysis of Randomized Controlled Trials

**DOI:** 10.3390/biomedicines13010177

**Published:** 2025-01-13

**Authors:** Elham Razmpoosh, Mala S. Sivanandy, Alan M. Ehrlich

**Affiliations:** 1Department of Health Research Methods, Evidence and Impact (HEI), McMaster University, Hamilton, ON L8S 4L8, Canada; erazmpoosh@mcmaster.ca; 2PCOS Center, Division of Endocrinology, Beth Israel Deaconess Medical Center, Harvard Medical School, Boston, MA 02215, USA; msivanan@bidmc.harvard.edu; 3Department of Family Medicine and Community Health, UMass Chan Medical School, Worcester, MA 01655, USA; 4EBSCO Information Services, Ipswich, MA 01938, USA

**Keywords:** gut microbiota, body mass index, inflammation, women’s health, endocrine disorders, data aggregation

## Abstract

**Background**/**Objectives**: This systematic review and meta-analysis aimed to investigate the effect of prebiotics, alone or as part of synbiotics, on cardiometabolic parameters in polycystic ovary syndrome (PCOS) women. **Methods**: Databases, including PubMed, Scopus, ISI Web of Science, Embase, and the Cochrane Central Register of Controlled Trials, were searched for relevant randomized-controlled trials (RCTs) until 12 December 2024. Changes in mean ± standard deviations were extracted and combined using a random-effects model. Bias was assessed using Cochrane risk of bias and evidence quality with GRADE. **Results**: Twenty RCTs with 1271 participants were included. Results showed high-quality evidence supporting prebiotics’ effects, alone or as part of synbiotics, in reducing body-mass index [*n* = 853; weighted-mean difference (WMD): −0.510, 95%CI: −0.669, −0.351 kg/m^2^] and diastolic blood pressure (WMD: −2.218, 95%CI: −4.425, −0.010 mmHg), moderate-quality evidence for weight, waist-to-hip ratio, and triglycerides improvements, and low or very-low-quality evidence for waist circumference (WC), fat mass, fasting plasma glucose, fasting insulin, low-density lipoprotein (LDL), total cholesterol (TC), high sensitive-C reactive protein, total testosterone, follicle-stimulating hormone and free androgen index improvements. Subgroup analyses revealed possible reduction in LDL with prebiotics, as well as possible decreases in WC, TC, and total testosterone with synbiotics. Dietary approaches to stop hypertension diet improved insulin sensitivity. **Conclusions**: This study suggests that prebiotics may beneficially affect several cardiometabolic parameters in PCOS women. Approximately one-third of the results were based on moderate-to-high-quality evidence. This study highlights the need for future well-designed, larger RCTs with longer treatment duration to strengthen the evidence base and guide clinical decision-making.

## 1. Introduction

Polycystic ovary syndrome (PCOS) is a major public health concern worldwide and is the most common endocrine disorder in women of childbearing age [1]. It is characterized by a myriad of clinical and laboratory abnormalities, including menstrual irregularities, hyperandrogenism, insulin resistance, and obesity. This syndrome poses long-term health risks, such as an increased susceptibility to type 2 diabetes mellitus, dyslipidemia, hypertension, metabolic syndrome, and cardiovascular disease [2]. Currently, there is no cure for PCOS. Apart from lifestyle and cosmetic measures, medications such as oral contraceptive pills, spironolactone, and metformin are currently being used to treat its diverse symptoms. Even combining all the above medications may not fully correct the multiple issues that some women with PCOS have. Therefore, clinicians are in constant search for newer therapies, especially ones that could alter the pathophysiology of PCOS and prevent developing its complications.

Dysbiosis of gut microbiota has been observed in individuals with PCOS, potentially contributing to chronic inflammation, insulin resistance, and metabolic disturbances [3]. The gut microbiome of women with PCOS exhibits lower alpha diversity [4] and reduced abundance of beneficial bacteria that produce short-chain fatty acids (SCFA) [5]. Serum levels of proinflammatory bacterial products such as lipopolysaccharide (LPS) are higher in women with PCOS compared to healthy controls [6].

Prebiotics are non-digestible, selectively fermentable dietary food/fiber sources that feed the gut microbiota, stimulate the growth of beneficial intestinal bacteria and improve host health [7]. Inulin, fructo-oligosaccharide (FOS), and galacto-oligosaccharides are a few examples of prebiotics. Prebiotic use is associated with a significant reduction in body weight in individuals with overweight or obesity [8]. In adults with baseline body mass index (BMI) ≥ 25 kg/m^2^, prebiotic supplementation has been shown to decrease total cholesterol and, in trials of patients with diabetes, increase HDL and reduce triglycerides (TG) [9]. All of the above are indicative of a plausible reversal of gut dysbiosis by optimizing the conditions for the beneficial bacteria to thrive and rebuild a healthy microbiome. Similarly, diets rich in fiber, like the Dietary Approaches to Stop Hypertension (DASH) diet, have been shown to positively influence gut microbiota [10]. Despite mounting evidence on the benefits of prebiotics and synbiotics (combination of prebiotics and probiotics) in various clinical contexts, their specific effects on women with PCOS remain an underexplored area.

Several prior systematic reviews and meta-analyses have primarily focused on investigating the influence of probiotics or synbiotics on PCOS, and they have yielded mixed results. A recent summary of past meta-analyses and systematic reviews examining the impacts of probiotics, synbiotics, and prebiotics in individuals with PCOS found that probiotics might offer potential benefits for specific PCOS-related measures like BMI, fasting plasma glucose (FPG), and lipid profiles. Conversely, synbiotics appeared less impactful on these parameters when compared to probiotics [11]. An umbrella review also explored the effects of pro-, pre-, and synbiotics on PCOS but included only two RCTs on prebiotics, which is inadequate for performing a reliable meta-analysis [12]. All the prior research in this domain has been notably lacking in investigating the specific effects of prebiotics within the realm of PCOS management.

We aimed to assess available randomized controlled trials (RCTs) and evaluate the overall effectiveness of prebiotics supplementation, either alone or as part of synbiotics, on cardiometabolic parameters in women diagnosed with PCOS.

## 2. Materials and Methods

### 2.1. Data Sources and Search Strategies

This systematic review and meta-analysis was conducted in compliance with the Cochrane Handbook for Systematic Reviews of Interventions and adhered to the Preferred Reporting Items for Systematic Reviews and Meta-Analyses (PRISMA) guidelines [13]. This study had been pre-registered on the International Prospective Register of Systematic Reviews (PROSPERO) with registration number CRD42023477930 on 14 November 2023.

### 2.2. Search Strategy

To find relevant studies on the effects of prebiotics, alone or as part of synbiotics, on cardiometabolic parameters in women with PCOS, we conducted a comprehensive literature search in electronic databases, including PubMed, Scopus, ISI Web of Science, Embase, and the Cochrane Central Register of Controlled Trials (CENTRAL), until 12 December 2024. Two sets of keywords, including the associated keywords for women with PCOS (the population type) and the keywords for prebiotics and synbiotics (the intervention type), were combined using the Boolean operator “OR” within each set and “AND” to combine them. The selection of keywords was based on the Medical Subject Headings (MeSH) database and other pertinent non-MeSH terms. The detailed search strategies are reported in Appendix A.

### 2.3. Study Selection Criteria and Outcomes and Prioritization

The study’s inclusion criteria consisted of RCTs with either parallel or crossover designs involving women aged 18 years or older who had been diagnosed with PCOS according to the Rotterdam criteria [14] and aimed to assess the impact of interventions involving prebiotics or synbiotics. Prebiotic interventions included substances such as inulin, psyllium, and FOS, as well as high-fiber diets [10] such as the DASH diet [15,16]. Synbiotic interventions involve the combination of prebiotics and probiotics. To be included, the trials also needed to report clinical or laboratory outcomes from amongst various parameters, including glycemic indices, lipid profile, hormonal parameters including follicle-stimulating hormone (FSH) as well as luteinizing hormone (LH), total testosterone, dehydroepiandrosterone sulfate (DHEAS), oxidative stress and inflammatory markers, blood pressure, and various anthropometric measurements like weight, BMI, waist circumference (WC), hip circumference (HC), waist-to-hip ratio (WHR), lean body mass and fat mass.

Animal studies, retrospective, prospective-cohort, cross-sectional, and pretest-post-test investigations were excluded. Studies that did not primarily or secondarily evaluate any of the previously mentioned outcomes of interest were also excluded. Articles that combined other drug interventions with prebiotics and synbiotics, studies lacking full texts, duplicate publications of results, and articles without control groups were also omitted. Furthermore, studies involving individuals without a definite diagnosis of PCOS, such as those with insulin resistance, infertility, or acne but without confirmed PCOS, were not considered. There were no language restrictions imposed. Notably, as outlined in the Cochrane Handbook for Systematic Reviews (MECIR Box 5.3.b), studies with multiple intervention arms require evaluating each arm individually, with only those meeting the eligibility criteria included in the analysis [13]. Two main reviewers (ER and MS) reviewed the titles and abstracts of articles according to the eligibility criteria. Any disagreements during the initial screening were resolved through discussion, with a final consensus agreement facilitated by a third reviewer (AE). Subsequently, relevant articles were retrieved for further evaluation, and duplicates were removed.

Primary outcomes were mean changes in anthropometric indices and changes in inflammatory and oxidative stress parameters. Secondary outcomes were changes in hormonal status, glycemic indices, lipid profile, and blood pressure.

### 2.4. Data Extraction

Two review authors, ER and MS, independently conducted data extraction from eligible studies. Apart from baseline information, they gathered various data points from each eligible article, including study design, types of intervention, types of prebiotics, study duration, measured outcomes, and mean baseline BMI of participants. Prebiotic interventions included utilizing inulin, psyllium, and FOS, as well as high-fiber diets such as the DASH diet, whereas synbiotic interventions involved the combination of prebiotics and probiotics. The mean changes in outcomes, along with their respective standard deviations (SD), were utilized for the final meta-analysis. We calculated the standard error (SE) from the available data and subsequently converted it to SD using established methodologies found in the Cochrane Handbook of Systematic Reviews [13] and methods described by Wan et al. [17]. Any uncertainties or discrepancies were resolved through group discussions involving a third reviewer, AE. When essential information was missing from the published report, attempts were made to contact the study authors to obtain the necessary details.

### 2.5. Assessment of Risk of Bias and Quality of Evidence

The evaluation of the risk of bias in individual studies was conducted using the revised Cochrane risk-of-bias tool for randomized trials (RoB2) [18], comprising five key domains: bias arising from the randomization process, bias due to deviations from intended interventions, bias related to missing outcome data, bias concerning outcome measurement, and bias linked to the selection of reported results. The final judgments and the overall risk of bias were categorized as either “low”, “high”, or expressed as having “some concerns”. In addition, the Grading of Recommendations Assessment, Development, and Evaluation (GRADE) approach was employed to appraise the overall quality of evidence. This evaluation encompassed domains such as risk of bias, publication bias, result imprecision, heterogeneity, and indirectness of evidence. Ultimately, the quality of evidence was classified as either “high”, “moderate”, “low”, or “very low” [19].

### 2.6. Statistical Analyses

The weighted mean difference (WMD) and its corresponding SE were either extracted or computed based on the mean differences between the intervention and control groups in each study and were combined using the DerSimonian and Laird random-effects model, which accounts for between-study variations. Heterogeneity between included studies was evaluated using Cochran’s *Q* test and the *I*-squared (*I*^2^) test.

Multiple subgroup analyses were performed to explore the origins of heterogeneity, including intervention duration, the specific types of interventions (prebiotics or synbiotics), the type of prebiotics used (fiber, FOS, or inulin), using DASH diet, use of low-calorie (LC) diet, the country of study, and baseline BMI categories. A summary of the effect and the corresponding assessment of heterogeneity were provided for each subgroup analysis.

Publication bias was evaluated using visual inspection of funnel plots, and any asymmetry was assessed using Begg’s adjusted rank correlation and Egger’s tests [20]. If publication bias was noted, Duval and Tweedie’s trim-and-fill method was employed to rectify funnel plot asymmetry [21]. Sensitivity analyses were conducted by systematically excluding one study or a group of studies at a time to ensure that the chosen values did not unduly influence the overall results. All statistical analyses were carried out using Stata (version 14.2; StataCorp). Two-sided *p*-values less than 0.05 were considered statistically significant.

## 3. Results

### 3.1. Flow of Studies

Figure 1 outlines the study selection process. Initially, we identified 2229 records, of which 689 duplicates were removed. After reviewing the titles and abstracts, eighty records were selected for full-text assessment. Among these, twenty studies were excluded due to their lack of relevance to the outcome measures or methodologies. Within the remaining articles, further exclusions were made for various reasons; four studies lacked fiber or other prebiotics in their dietary interventions [22,23,24,25]; six focused solely on probiotics [26,27,28,29,30]; one study enrolled women with acne without specifying PCOS criteria [31]; one study lacked randomization [32]; seven employed inositol [33,34,35,36,37,38,39], and three incorporated metformin [40,41] or acarbose [42] in conjunction with dietary interventions. One study used a high-fiber diet in both the intervention and control groups [43], and another study did not use a high-fiber diet as the intervention [44].

One study was a protocol with no published results [45]. Three studies used the Mediterranean diet as the intervention [46,47] or another nutritional intervention [48] but lacked data to ascertain if these were high-fiber diets. One study used a combination of the DASH diet and curcumin [49]. Additionally, two studies initially existed as abstracts [50,51], later published in full texts [52,53]. Five studies lacked a control group [54,55,56,57,58], and two studies reported similar results [59,60], which were published previously. Finally, twenty articles were included in this systematic review and meta-analysis [52,53,61,62,63,64,65,66,67,68,69,70,71,72,73,74,75,76,77,78]. Among these, six were secondary analyses of prior studies [52,66,68,70,77,78]. Since these secondary analyses reported new outcomes that aligned with the inclusion criteria, we incorporated them into the present study. Notably, in the studies by Esmaeilinezhad et al. [65,66], four distinct groups were included (two different interventions with two corresponding controls, all meeting the inclusion criteria), resulting in separate analyses of the two-by-two intervention and control groups.

### 3.2. Risk of Bias and Quality of Evidence

According to the RoB2, eleven studies were classified as having a “low” risk of bias [53,66,68,69,70,71,73,74,76,77,78], six articles as “some concerns” [52,61,63,64,65,67], and three studies with a “high” risk of bias [62,72,75]. Sufficient details on allocation concealment and randomization were not reported for six studies [52,61,62,63,75]. Based on “the risk of bias due to deviations from the intended interventions”, three records represented a “some concerns” risk of bias [62,65,75], and only one article reported a “high” risk of bias [72]. Moreover, three articles showed a “some concerns” risk of bias after assessing the risk of bias based on the “measurement of the outcome” [62,72,75]. In the assessment of the risk of bias based on the “selection of the reported result”, one study had a “some concerns” risk of bias [67] (Appendix A). The summary of the risk of bias in the included studies is shown in Figure 2. Considering the GRADE quality of evidence, fat mass, low-density lipoprotein cholesterol (LDL-C), total cholesterol (TC), FPG, insulin, Quantitative Insulin Sensitivity Check Index (QUICKI), total testosterone, and malondialdehyde (MDA) had “low” GRADE quality. “Very-low” quality applied to HC, WC, high-density lipoprotein cholesterol (HDL-C), Homeostatic Model Assessment for Insulin Resistance (HOMA-IR), FSH, LH, DHEAS, free androgen index (FAI), high-sensitivity C-reactive protein (hs-CRP), nitric oxide (NO), and total antioxidant capacity (TAC). “Moderate” quality was noted for weight, WHR, TG, and systolic blood pressure (SBP), while BMI and diastolic blood pressure (DBP) had a “high” quality of evidence (Appendix A).

### 3.3. Characteristics of Included Studies

The characteristics of the included studies are provided in Table 1. All had a parallel design and used Rotterdam criteria for diagnosing the PCOS among participants [14]. Eighteen studies were conducted in Iran, one in Poland [63] and one in China (2023) [75]. Participants’ mean age ranged between 22.1 to 32.1 years. Participants’ baseline BMI was 25–29.9 kg/m^2^ in thirteen studies [64,65,66,68,69,71,72,73,74,75,76,77,78] and 30–34.4 kg/m^2^ in seven studies. The intervention duration was 8 weeks in seven studies [52,61,64,65,66,72,75] and 12 weeks in thirteen studies [53,62,63,67,68,69,70,71,73,74,76,77,78]. Ten studies used prebiotics [52,61,62,67,68,69,72,74,75,78], and ten studies used synbiotics [53,63,64,65,66,70,71,73,76,77]. Twelve studies used inulin or FOS [53,63,64,65,66,70,71,73,74,76,77,78], five studies used fiber [52,61,62,67,75], one study used psyllium [72], and two studies used dextrin as prebiotics [68,69]. Four studies used a combination of LC and DASH diet as the intervention [52,61,62,67], one study used LC and high fiber [75], and one used LC diet along with synbiotics as the intervention [63].

### 3.4. Meta-Analysis

#### 3.4.1. Primary Outcomes

**Anthropometric parameters [body mass index (BMI), body weight, waist circumference (WC), hip circumference, waist-to-hip ratio (WHR), fat mass].** Reduction in BMI was demonstrated in women diagnosed with PCOS with overweight or obesity when treated with prebiotics, either alone or as part of synbiotics, in fourteen RCTs that included 853 participants [61,62,63,64,65,67,69,70,71,72,73,74,75,77] (Figure 3) (WMD: −0.510, 95%CI: −0.669, −0.351 kg/m^2^; *p* < 0.001; *I*^2^ = 0.0%). Reduction in body weight was demonstrated in fourteen RCTs that included 852 participants [61,62,63,64,65,67,69,70,71,72,73,74,75,77] (WMD: −1.857, 95%CI: −2.464, −2.249 kg; *p* < 0.001; *I*^2^ = 9.8%). A decrease in WC was demonstrated in nine RCTs with 565 participants [52,62,63,64,65,69,70,77,78] (WMD: −3.11, 95%CI: −4.193, −2.028 cm; *p* < 0.001; *I*^2^ = 69.8%). No change in HC was noted in eight RCTs that included 515 participants [52,62,63,64,65,69,70,77] (WMD: −0.565, 95%CI: −1.426, 0.297 cm; *p* = 0.199; *I*^2^ = 64.2%). Reduction in WHR was noted in six RCTs that included 401 participants [62,63,64,65,70,72] (WMD: −0.015, 95%CI: −0.026, −0.003; *p* = 0.011; *I*^2^ = 49.5%). Reduction in fat mass was demonstrated in four RCTs with 202 participants [62,63,72,75] (WMD: −1.496, 95%CI: −2.205, −0.787; *p* < 0.001; *I*^2^ = 62.2%). Details of the meta-analysis results for anthropometric indices are presented in Appendix A, and corresponding plots are provided in Appendix A. Subgroup analyses revealed reduced WC and WHR in individuals with a baseline BMI of 25–29.9 kg/m^2^. Additionally, synbiotics and inulin, as the types of prebiotics, were associated with WC reduction.

**Inflammatory and oxidative stress markers [high-sensitivity c-reactive protein (hs-CRP), total antioxidant capacity (TAC), malondialdehyde (MDA), and nitric oxide (NO)].** Reduction in hs-CRP was demonstrated in seven RCTs that involved 466 participants [52,53,66,68,71,74,76] (WMD: −0.594, 95%CI: −0.968, −0.221 mg/L; *p* = 0.002; *I*^2^ = 96.3%). Subgroup analyses indicate potential hs-CRP reduction with psyllium and fiber prebiotics (3 studies, *n* = 296; WMD: −1.434; −1.642, −1.227 mg/L; *p* < 0.001; *I*^2^ = 0.0%), and in participants with baseline BMI ≥ 30 kg/m^2^ (2 studies, *n* = 128; WMD: −1.433, 95%CI: −1.642, −1.225 mg/L; *p* < 0.001; *I*^2^ = 0.0%). Meta-regression suggests age (β = 0.189, *p* < 0.001) as a significant predictor of hs-CRP changes. No effect on TAC was noted in three RCTs that included 195 participants [61,66,71] (WMD:135.935; 95%CI: −3.201, 275.070 mmol/L; *p* = 0.056; *I*^2^ = 77.3%). No change with MDA was noted in three RCTs that included 212 participants [65,67,71] (WMD: −0.095; 95%CI: −0.510, 0.321 μmol/L; *p* = 0.654; *I*^2^ = 87.2%). An increase in NO was noted in three RCTs that included 170 participants [67,71,74] (WMD:5.372, 95%CI: 1.131, 9.613 μmol/L; *p* = 0.013; *I*^2^ = 87.2%). The meta-analysis outcomes for inflammatory and oxidative stress markers are summarized in Appendix A, with corresponding plots in Appendix A.

#### 3.4.2. Secondary Outcomes

**Lipid profile [total cholesterol (TC), low-density lipoprotein cholesterol (LDL), high-density lipoprotein cholesterol (HDL), triglycerides (TG)].** A decrease in TC was demonstrated in ten studies that included 621 participants [61,63,64,66,68,70,72,73,76,78] (WMD: −8.563, 95%CI: −13.720, −3.406 mg/dL; *p* = 0.001; *I*^2^ = 40.8%). In a sensitivity analysis, excluding the study by Gholizadeh Shamasbi et al. (2019) [68], all evidence of heterogeneity disappeared (WMD: −6.253, 95%CI: −8.917, −3.590 mg/dL; *p* < 0.001; *I*^2^ = 0.0%). Subgroup analyses showed consistent TC reductions with synbiotics (6 studies, *n* = 413; WMD: −5.730, 95%CI: −9.775, −1.685 mg/dL; *p* = 0.005; *I*^2^ = 11%) as well as in participants with baseline BMI ≥ 30 kg/m^2^ and an 8-week intervention duration (4 studies, *n* = 251; WMD: −11.248, 95%CI: −20.024, −2.472 mg/dL; *p* = 0.012; *I*^2^ = 0.0%).

Reduction in LDL was demonstrated in ten studies with 621 participants [61,63,64,66,68,70,72,73,76,78] (WMD: −10.149, 95%CI: −17.168, −3.129 mg/dL; *p* = 0.005; *I*^2^ = 72.2%). Subgroup analyses revealed consistent reductions, especially with prebiotics like psyllium or fiber (3 studies, *n* = 158; WMD: −30.691,95%CI: −40.862, −20.519 mg/dL; *p* < 0.001; *I*^2^ = 0.0%). LDL levels decreased with prebiotics and synbiotics interventions over 8 weeks (4 studies, *n* = 251; WMD: −20.160, 95%CI: −30.452, −9.868 mg/dL; *p* < 0.001; *I*^2^= 6.4%). A sensitivity analysis, removing the study by Gholizadeh Shamasbi et al. (2019) [68], reduced *I*^2^ from 75.4% to 52.1% (WMD: −6.247, 95%CI: −11.771, −0.722 mg/dL; *p* = 0.027; *I*^2^ = 52.11%). Variations in participants’ baseline BMI were a significant source of heterogeneity.

Reduction in TG (Figure 4) was demonstrated in ten RCTs with 621 participants [61,63,64,66,68,70,72,73,76,78](WMD: −13.974, 95%CI: −22.459, −5.490 mg/dL; *p* = 0.001; *I*^2^ = 33.8%).

No effect on HDL was seen in ten RCTs that included 621 participants [61,63,64,66,68,70,72,73,76,78] (WMD: 2.152; 95%CI: −0.028, 4.277 mg/dL; *p* = 0.057; *I*^2^ = 59.9%). A sensitivity analysis, omitting the study by Gholizadeh Shamasbi et al. (2019), [68] reduced *I*^2^ from 63.5% to 46.3% (WMD: 1.418, 95%CI: −0.515, 3.352 mg/dL; *p* = 0.151; *I*^2^ = 46.3%).

The meta-analysis results for lipid profile parameters are in Appendix A, with plots in Appendix A.

**Glycemic indices [fasting plasma glucose (FPG), fasting insulin, quantitative insulin sensitivity check index (QUICKI), homeostasis model assessment for insulin resistance (HOMA-IR)].** A reduction in FPG level was noted in eleven RCTs with 686 participants [53,61,63,64,65,67,68,72,73,75,78] (WMD: −3.942, 95%CI: −6.810, −1.074 mg/dL; *p* = 0.007; *I*^2^ = 93.7%). Subsequent subgroup analyses pinpointed potential sources of this heterogeneity, including the implementation of a LC diet, and intervention duration. Reduction in fasting insulin was demonstrated in ten RCTs that involved 624 participants [53,61,63,64,65,67,72,73,75,78] (WMD: −2.816, 95%CI: −4.203, −1.429 mg/dL; *p* < 0.001; *I*^2^ = 83.7%). Subgroup analyses showed consistent results in studies using the DASH diet intervention (2 studies, *n*= 108; WMD: −4.679, 95%CI: −5.521, −3.837 mg/dL; *p* < 0.001; *I*^2^ = 0.0%). The study location was identified as another source of heterogeneity. An increase in QUICKI level (insulin sensitivity) was noted in seven RCTs that included 457 participants [53,61,65,67,72,73,78] (WMD:0.016, 95%CI: 0.007, 0.026; *p* = 0.001; *I*^2^ = 91.9%). Subgroup analyses showed an increase in QUICKI in studies lasting 12 weeks. No change in HOMA-IR was noted in nine studies [53,61,64,65,67,72,73,75,78] (WMD: −0.417, 95%CI: −0.971, 0.137; *p* = 0.140; *I*^2^ = 93.0%). Using the DASH diet was the source of between-study heterogeneity (Appendix A).

### 3.5. Blood Pressure [Systolic Blood Pressure (SBP) and Diastolic Blood Pressure (DBP)]

No change in SBP was noted in three RCTs with 236 participants (WMD: −1.302; 95%CI: −4.668,2.064 mmHg; *p* = 0.448; *I*^2^ = 31.4%). However, a decrease in DBP was noted [66,70,74] (WMD: −2.218, 95%CI: −4.425, −0.010 mmHg; *p* = 0.049; *I*^2^ = 0.0%) (Appendix A).

**Hormonal parameters [total testosterone, sex hormone-binding globulin (SHBG), free androgen index (FAI), dehydroepiandrosterone sulfate (DHEAS), follicle-stimulating hormone (FSH) and luteinizing hormone (LH)].** A reduction in total testosterone level was noted in seven RCTs [62,63,65,67,71,75,78] (WMD: −0.268, 95%CI: −0.473, −0.063 IU/L; *p* = 0.01; *I*^2^ = 80.04%). Subgroup analyses suggest a potential decrease in total testosterone in studies with synbiotics or inulin as interventions. Similar trends were seen in participants with baseline BMI ≥ 30 kg/m^2^. No meaningful change in SHBG was noted in six RCTs that included 324 participants [62,63,67,71,75,78] (WMD: 13.733, 95%CI: −0.094, 27.560 IU/L; *p* = 0.0520; *I*^2^= 97.8%). Reduction in FAI was demonstrated in five RCTs that included 285 participants [62,67,71,75,78] (WMD: −0.227; 95%CI: −0.305, −0.148 IU/L; *p* < 0.00; *I*^2^ = 60.2%). Subgroup analyses revealed a decrease in FAI in studies conducted in Iran. No change in DHEAS was noted in three RCTs with 161 participants [63,68,71] (WMD: 0.179, 95%CI: −0.381, 0.739 IU/L; *p* = 0.531; *I*^2^ = 65.3%). The sources of between-study heterogeneity were not found. An increase in FSH was demonstrated in four RCTs that included 191 participants (WMD: 1.020; 95%CI: 0.475, 1.564 IU/L; *p* < 0.001; *I*^2^ = 6.7%), but no effects on LH levels were observed [33,63,65,67] (WMD: 1.050, 95%CI: −0.392, 2.493 IU/L; *p* = 0.153; *I*^2^ = 66.5%). No change in hirsutism was noted in three RCTs with 149 participants [63,71,78] (WMD: −1.128, 95%CI: −2.286, 0.03; *p* = 0.056; *I*^2^ = 90.8%). The meta-analysis results for hormonal levels are outlined in Appendix A, with corresponding plots in Appendix A.

### 3.6. Outcomes Not Included in the Meta-Analysis

One study reported hemoglobin A1C with no significant changes [53]; another study reported a significant decrease in the anti-müllerian hormone [67].

### 3.7. Publication Bias and Sensitivity Analyses

Begg’s tests suggested publication bias for the effects of prebiotics and synbiotics on HC measures (Begg’s *p*-value = 0.02, Egger’s *p*-value = 0.49), and subsequent trimming of the data led to the inclusion of eight studies in the analysis. Egger’s tests suggested publication bias for WC (Begg’s *p*-value = 0.788, Egger’s *p*-value = 0.013), FSH (Begg’s *p*-value = 0.624, Egger’s *p*-value = 0.01), and HDL. Trimming data included eight and five studies for these two parameters, respectively.

An influence analysis was conducted for each outcome, and the findings revealed that excluding any individual study had no impact on the result, except for total testosterone and NO measures. Nasri et al. (2018) [71] influenced the overall analysis of NO. A sensitivity analysis was conducted, excluding three studies [53,64,70] without reported prebiotic dosage, which did not impact heterogeneity. Another analysis excluded Pourbei et al. [72] due to reported metformin use, which was adjusted between groups, and no significant changes in heterogeneity were found. Additionally, a sensitivity analysis was performed, excluding articles of very low and low quality based on the GRADE evaluation. The results revealed no significant changes in any of the parameters.

## 4. Discussion

This systematic review and meta-analysis of RCTs comprehensively evaluated the impact of prebiotics, alone or as part of synbiotics, on cardiometabolic parameters in adult women with PCOS. The present study showcased the possible effectiveness of this intervention in improving PCOS-related outcomes.

The findings highlighted evidence for the efficacy of 8–12 weeks of treatment with prebiotics, alone or as part of synbiotics, in reducing BMI and DBP with high-quality evidence, possible improvement in weight, WHR, and TG, with moderate-quality evidence, and possible advantages in other parameters with low or very-low quality of evidence, including WC, fat mass, FPG, fasting insulin, LDL, TC, hs-CRP, and total testosterone, FSH, and FAI. Conversely, no statistically significant changes emerged in other parameters, encompassing SBP, HC, HOMA-IR, HDL, LH, SHBG, and DHEAS, as well as TAC and MDA.

The inherent capacity of prebiotics to fuel the growth of existing bacteria and enhance SCFA production is likely pivotal in promoting gut health, regardless of probiotic supplementation [7,79]. The positive overall impacts of prebiotics on inflammatory and oxidative stress markers might be due to multiple factors, including the reduction in the firmicutes population, modulation of gut microflora composition, and regulation of immune responses. These actions collectively may help mitigate systemic inflammation and bolster antioxidant factors [80].

The observed enhancements in anthropometric indices among PCOS patients following prebiotics or synbiotics supplementation may be linked to alterations in gut flora, stimulating beneficial gut bacteria and positively impacting energy balance [81,82]. Additionally, prebiotics may induce weight loss by reducing serum ghrelin levels and elevating hormones like leptin, glucagon-like peptide−1 and −2, as well as peptide YY [83]. The above might lead to decreased appetite, improved insulin signaling in adipose tissue, and reduced fat accumulation.

Rates of overweight or obesity in women with PCOS of 38–88% have been reported [84], and this can result in worsening insulin resistance or other metabolic complications. Evidence has shown that modest weight reduction, starting at about 2–5% of body weight, can improve menstrual irregularities and fertility in patients with PCOS and infertility [85]. Moreover, a cohort study found that for individuals at risk of type−2 diabetes, every kilogram of weight loss correlated with a 16% decrease in the risk of progressing to type−2 diabetes [85,86]. Therefore, it is crucial to consider clinical relevance alongside statistical significance when evaluating interventional outcomes. For instance, given the 12% higher risk of type−2 diabetes among women with overweight or obesity who have PCOS [87], and considering our finding of an average 1.857kg decrease in body weight (and a −0.510 kg/m^2^ decrease in BMI) following prebiotic or synbiotic use, even this modest reduction could yield significant benefits based on existing evidence.

In a meta-analysis of thirty-five studies, a high prevalence of central obesity with a relative risk of 1.73 was observed in women with PCOS [88]. In addition, a cohort study reported that a 5 cm reduction in WC has been linked to a 9% decrease in mortality over 6.7 years, regardless of BMI in healthy individuals [89]. Given the present finding of a 3.11 cm decrease in WC after pre- and synbiotic intervention in PCOS patients within 12 weeks, there is the potential for a clinically significant impact if these effects were sustained over a longer period.

This meta-analysis also indicated a high level of evidence for reducing DBP in populations with PCOS. The potential mechanisms through which synbiotics may positively impact DBP include slowing down carbohydrate digestion and glucose absorption, regulating blood sugar and improving insulin sensitivity, which in turn impacts blood pressure control [90]. Prebiotics also improve lipid profiles, mitigating arterial stiffness and endothelial dysfunction positively affecting DBP. Lastly, prebiotics aid weight management by promoting satiety and reducing calorie intake, potentially lowering body weight. Since obesity is a major risk factor for hypertension, weight loss facilitated by prebiotics could contribute to lowering DBP [91]. However, given the limited number of studies and the marginal statistical significance of 0.049, further high-quality research is needed to thoroughly investigate the impacts of pre- and synbiotics on blood pressure.

Findings from subgroup analyses also revealed that prebiotic use showed possible reductions in LDL levels, and synbiotic interventions exhibited a decrease in WC, TC, TG, and total testosterone. The present study also found that studies using inulin or FOS demonstrated possible reductions in WC and total testosterone, while those with other types of prebiotics, including fiber or psyllium, displayed possible decreased LDL and hs-CRP levels. Evidence indicates that a fiber-rich diet induces changes in gut microbiota, leading to increased circulating levels of GLP−1 and the production of SCFAs like acetate and butyrate in the colon. This is especially so in individuals with hyperinsulinemia [92]. Similar observations have been made in studies focusing on the effects of prebiotics or carbohydrates with prebiotic properties in overweight individuals, conducted over durations of 4 to 12 weeks [92,93]. These findings also align with our analysis that the DASH diet, high in fiber with prebiotic properties, notably improves insulin sensitivity in participants with PCOS during an overall intervention of 8–12 weeks.

Regarding different findings based on different prebiotics types, it should be considered that certain types of fiber might have a more substantial impact on reshaping the gut microbiome and influencing metabolic health in individuals with PCOS, compared to the more targeted effects of inulin or FOS on specific beneficial bacteria [7]. Further research focusing specifically on the effects of different types of prebiotics and dietary fiber in populations with PCOS could provide deeper insights into their impacts on metabolic health markers.

Synbiotic supplements can positively affect total testosterone by improving intestinal function, reducing harmful bacteria, and lowering metabolic endotoxemia, thus supporting normal ovarian function and decreasing androgen production [64]. The beneficial effect of synbiotics on TC and TG levels might stem from probiotics within synbiotics incorporating cholesterol into intestinal bacteria cell walls, hindering its absorption [53].

The subgroup analyses revealed that trials with participants having a mean baseline BMI ≥ 30 kg/m^2^ potentially experienced decreases in TC, hs-CRP, and total testosterone. Conversely, studies involving PCOS participants with an average baseline BMI of 25–29.9 kg/m^2^ showed potential reductions in WC and WHR. These outcome variations may result from differing metabolic profiles and body compositions within these BMI ranges. Trials with participants having a mean baseline BMI ≥ 30 kg/m^2^ likely include individuals with more pronounced metabolic dysregulation, such as higher insulin resistance and dyslipidemia, often associated with increased central adiposity [94]. Conversely, studies with participants falling between 25–29.9 kg/m^2^ might indicate more focused changes in regional adiposity and body composition due to the less severe metabolic derangements in this range. These differences underline the need for tailored interventions considering the diverse metabolic profiles within the PCOS population across different BMI categories.

The present data suggests a possible increase in insulin sensitivity over 12 weeks. However, rectifying underlying metabolic dysregulations in PCOS, essential for lasting improvements, typically requires longer interventions [95]. Achieving substantial and enduring enhancements in insulin sensitivity often demands extended periods due to the comprehensive metabolic changes involved.

Findings related to HDL, LDL, and TC were significantly impacted by the study authored by Gholizadeh-Shamasbi et al. [68]. Gholizadeh-shamasbi et al. utilized resistant dextrin as a prebiotic intervention, setting their study apart. However, the sensitivity analysis showed that not including this outlier did not affect the result in a significant way.

The present study exhibited several strengths. It represented the inaugural systematic review and meta-analysis examining the impact of pre- and synbiotics, considered as sources of prebiotics, on multiple clinical outcomes in PCOS patients. The present study meticulously incorporated all RCTs exploring the effects of prebiotics, synbiotics, and high-fiber interventions potentially acting as prebiotics. Furthermore, the GRADE tool was utilized to assess the certainty of evidence, enhancing the clarity and validity of the findings, as it is a recognized and reliable method for this purpose.

The present study’s limitations should be noted. All RCTs focused on women with BMI ≥ 25 kg/m^2^, so the applicability of these findings may not extend to women with PCOS women who have a normal BMI. Eighteen of the twenty primary RCTs were conducted in Iran, so the applicability of these findings beyond that region may be restricted. Additionally, the present meta-analysis observed significant variation among the studies in specific analyses, which lowers the confidence of the results. Furthermore, we included studies utilizing high dietary fiber interventions, such as the DASH diet, which exceeds 20 g per day. However, there was not enough data to do any further analyses concerning fiber dosage. Finally, only five out of fifteen findings that were statistically significant were supported by moderate or high-quality evidence according to the GRADE criteria, indicating the potential of additional unidentified biases affecting many of the results.

## 5. Conclusions

In conclusion, there is a need for a multi-organ, multifaceted approach to PCOS to build better combination therapies to personalize treatment and effectively treat PCOS and its metabolic complications. This comprehensive inaugural systematic review and meta-analysis of RCTs demonstrated that prebiotics, alone or as part of synbiotics, may improve cardiometabolic parameters by improving BMI, WC, WHR, body weight, fat mass, TG, and DBP in adult PCOS women with overweight or obesity. Also, this study found evidence for the possible improvements in glycemic, lipid, hormonal, inflammatory, and antioxidant (FPG, fasting insulin, LDL, TC, TG, hs-CRP, insulin sensitivity, NO and total testosterone) markers. While the study presents valuable insights, there is a need for large, well-designed RCTs with different types and doses of prebiotics on various phenotypes of women with PCOS over a longer period to strengthen the evidence base and guide clinical recommendations.

## Figures and Tables

**Figure 1 biomedicines-13-00177-f001:**
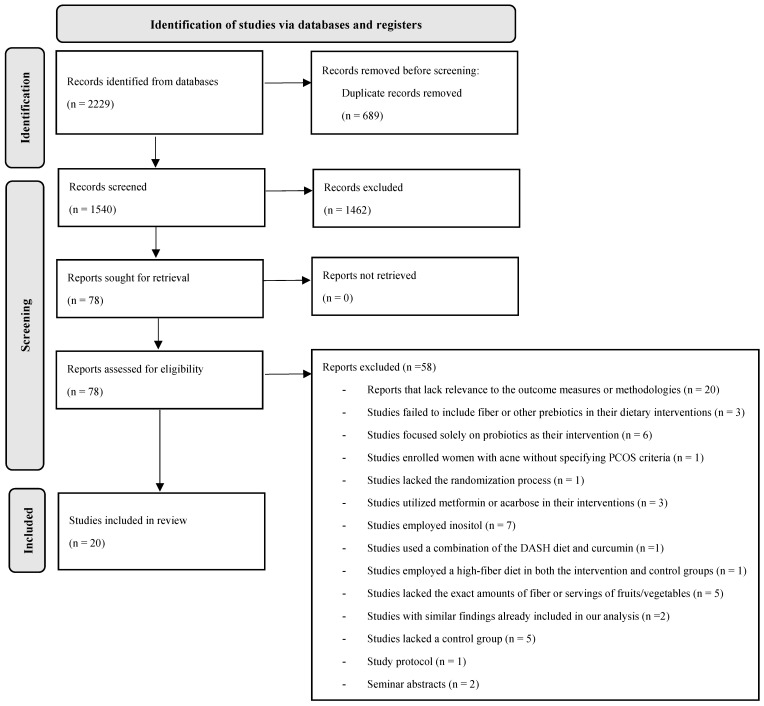
Study flow chart.

**Figure 2 biomedicines-13-00177-f002:**
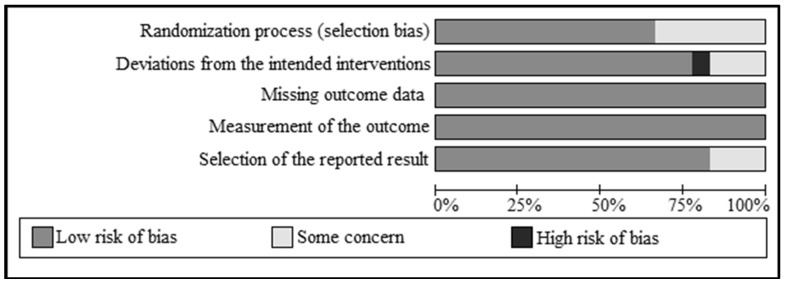
Summary of risk of bias of the included studies.

**Figure 3 biomedicines-13-00177-f003:**
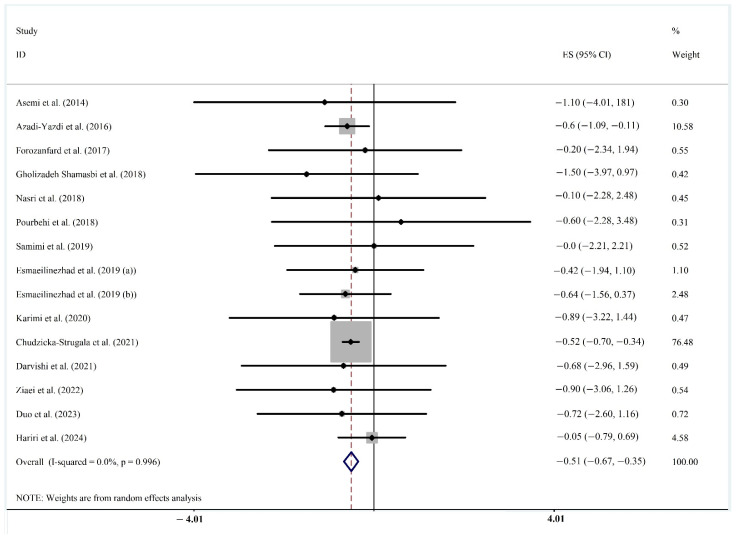
Forest plot of fourteen randomized controlled trial studies [61,62,63,64,65,66,67,69,70,71,72,73,74,75,77] showing weighted mean differences in “Body Mass Index” change (in kg/m^2^) between prebiotics and synbiotics intervention and control groups for all eligible studies. Analysis was conducted using a random-effects model. Squares depict the weight assigned to the corresponding study; the diamond represents the summary effect. ES, effect size.

**Figure 4 biomedicines-13-00177-f004:**
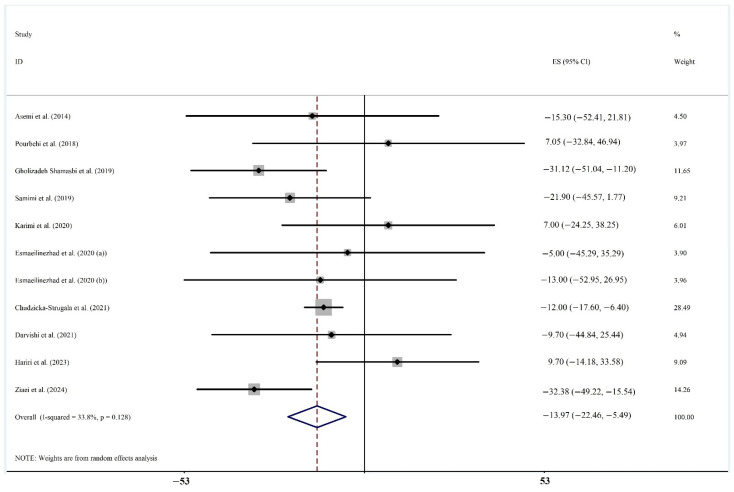
Forest plot of ten randomized controlled trial studies [61,63,64,65,66,68,70,72,73,76,78] showing weighted mean differences in “Triglycerides” change (in mg/dL) between prebiotics and synbiotics intervention and control groups for all eligible studies. Analysis was conducted using a random-effects model. Squares depict the weight assigned to the corresponding study; the diamond represents the summary effect. ES, effect size.

**Table 1 biomedicines-13-00177-t001:** Characteristic of included studies evaluating the effect of prebiotics and synbiotics on clinical and laboratory outcomes in women with polycystic ovary syndrome.

Reference	Number of Randomized Participants (No. in Intervention and Control Group)	Age, Mean (Intervention, Control) (y)	Country	Outcome	Type of Intervention	Type of Prebiotics	Duration (wk)	Results
[61]	48 (24, 24)	22.10, 24.70	Iran	Anthropometric indices, lipid profile, Oxidative stress markers	LC diet and DASH diet	Fiber	8	A significant decrease in TG, VLDL-C, weight, and BMI A Significant increase in TAC No significant changes in HDL-C, LDL-C, TC/HDL-C
[52]	48 (24, 24)	30.70, 29.40	Iran	Anthropometric indices, Glycemic parameters, inflammatory markers	LC diet and DASH diet	Fiber	8	A significant decrease in insulin, HOMA-IR, hs-CRP, WC, HC No significant changes in FPG
[62]	55 (28, 27)	32.13, 31.78	Iran	Hormonal parameters, Anthropometric indices	LC diet and DASH diet	Fiber	12	A significant decrease in weight, BMI, WC, HC, Fat mass, Lean mass, Androstenedione, A significant increase in SHBG, DPPH scavenging activity No significant changes in WHR, total testosterone, FAI
[63]	39 (20, 19)	30.80, 29.10	Poland	Lipid profile, Glycemic indices, anthropometric indices, Hormonal parameters	Synbiotics and LC diet and exercise	Inulin and FOS	12	A significant decrease in weight, BMI, WC, total testosterone No significant changes in fat%, WHR, HC, tight, hirsutism, ovarian volume, DHEAS, SHBG, LH, FSH, LH/FSH, insulin, FPG, ISI, TCLDL-C, HDL-C, TG
[64]	68 (34, 34)	30.40, 28.60	Iran	Lipid profile, Glycemic indices, anthropometric indices	Synbiotics	FOS	8	A significant decrease in FPG, insulin, HOMA-IR, Weight, BMI, WC, HC A significant increase in HDL-C No significant changes in TC, TG, LDL-C, Apelin, WHR
[75]	60 (30, 30)	32.00, 31	China	Hormonal parameters, Glycemic indices, anthropometric indices	LC diet and High fiber diet and exercise	Fiber	8	A significant decrease in Weight, BMI, Fat mass, Body fat%, VFA, FFM, and BMR, FPG, Insulin A significant increase in SHBG No significant changes in HOMA-IR, total testosterone, FAI
[65]	46 (23, 23)	30.04, 29.3	Iran	Hormonal parameters, Glycemic indices, Anthropometric indices, Blood pressure	Synbiotics	Inulin	8	A significant decrease in Weight, BMI, WC, WHR, FPG, Insulin, HOMA-IR, Total testosterone A significant increase in QUICKI No significant changes in HC, LH, FSH, LH/FSH
[66]	46 (23, 23)	29.52, 30.6	Iran	Lipid profile Inflammatory markers Oxidative stress markers Blood pressure	Synbiotics	Inulin	8	A significant decrease in SBP, DBP, hs-CRP, MDA, TC, TG, LDL-C/HDL-C A significant increase in TAC, HDL-C
[67]	60 (30, 30)	27.10, 25.6	Iran	Anthropometric indices Glycemic parameters Hormonal parameters Oxidative stress markers	LC diet and DASH diet	Fiber	12	A significant decrease in FPG, insulin HOMA-IR, AMH, SHBG, FAI, MDA A significant increase in QUICKI, NO No significant changes in FPG, Total testosterone, FSH, LH, 17-OH progesterone,
[68]	62 (31, 31)	28, 26	Iran	Anthropometric indices	Prebiotics	resistant Dextrin, (polysaccharide from corn, wheat, and other edible starches with fibrous properties)	12	A significant decrease in BMI, weight, HC, WC
[69]	62 (31, 31)	28, 26	Iran	Lipid profile, Glycemic indices, Hormonal parameters	Prebiotics	resistant Dextrin, (polysaccharide from corn, wheat, and other edible starches with fibrous properties)	12	A significant decrease in LDL-C, TC, TG, FPG, hs-CRP, DHEA-S, Free testosterone, Interval between menstrual cycles, and hirsutism score A significant increase in HDL-C No significant changes in Duration of menstrual bleeding
[76]	60 (34, 26)	28.14, 28.38	Iran	Lipid profile, inflammatory marker and atherogenicity indexes	Synbiotics	FOS	12	A significant decrease in hs-CRP No significant changes in TC, TG, LDL-C, HDL-C, Castelli risk index I, Castelli risk index II, Atherogenic coefficient, cholesterol index, TG/HDL-C
[70]	99 (50, 49)	28.5, 29	Iran	Anthropometric indices, Lipid profile, Blood pressure	Synbiotics	Inulin (FOS)	12	A significant increase in HDL-C A significant decrease in LDL-C, DBP No significant changes in weight, BMI, WC, HC, WHR, SBP, TC, TG
[53]	99 (50, 49)	28.5, 29	Iran	Glycemic indices, Inflammatory parameters,	Synbiotics	Inulin (FOS)	12	A significant increase in hs-CRP and Apelin-36 No significant changes in FPG, Glycohaemoglobin, Insulin, HOMA-IR, QUICKI
[71]	60 (30, 30)	25.2, 25.7	Iran	Inflammatory markers, Oxidative parameters, Hormonal indices, Anthropometric parameters	Synbiotics	Inulin	12	A significant decrease in mF-G scores, FAI, hs-CRP, MDA, A significant increase in HSBG, NO No significant changes in total testosterone, DHEAS, TAC, GSH
[72]	48 (24, 24)	27.7, 27.3	Iran	Anthropometric indices, Lipid profile, Glycemic indices	Prebiotics	psyllium)	8	A significant decrease in FM No significant changes in weight, BMI, FFM, WHR, SMM, insulin, TC, TG, LDL-C, HDL-C, HOMA-IR, QUICKI
[73]	60 (30, 30)	27.3, 27	Iran	Anthropometric indices, Lipid profile, Glycemic indices	Synbiotics	Inulin	12	A significant decrease in Insulin, HOMA-IR, TG, VLDL-C, FPG A significant increase in QUICKI No significant changes in TC, LDL-C, HDL-C, AIP
[74]	50 (25, 25)	29.4, 28.8	Iran	Anthropometric indices, Inflammatory markers, Oxidative stress markers, Blood pressure	Prebiotics	Inulin	12	A significant decrease in hs-CRP No significant changes in SBP, DBP, TOS, NO, Emdothelin-1
[77]	52 (28, 24)	28.42, 32.75	Iran	Anthropometric indices	Synbiotics	FOS	12	No significant changes in weight, BMI, WC and HC
[78]	50 (25, 25)	29.4, 28.8	Iran	Glycemic indices, Lipid profile, Hormonal Indices, WC	Prebiotics	Inulin	12	A significant reduction in insulin, HOMA-IR, TG, total testosterone, FAI, hirsutism and WC A significant increase in QUICKI and SHBG No significant changes in FPG, LDL HDL, and TC

Abbreviations: VLDL-C, very low-density lipoprotein cholesterol; BMI, body mass index; TAC, total antioxidant capacity; HDL-C, high-density lipoprotein cholesterol; LDL-C, low-density lipoprotein cholesterol; TC, total cholesterol; HOMA-IR, homeostatic model assessment for insulin resistance; hs-CRP, high-sensitivity c-reactive protein; WC, waist circumference; HC, hip circumference; FPG, fasting plasma glucose; SHBG, sex hormone-binding globulin; DPPH, 2,2-diphenyl-1-picrylhydrazyl; WHR, waist-to-hip ratio; FAI, free androgen index; DHEAS, dehydroepiandrosterone sulfate; LH, luteinizing hormone; FSH, follicle-stimulating hormone; ISI, insulin sensitivity index; VFA, visceral fat area; FFM, fat-free mass; BMR, basal metabolic rate; QUICKI, quantitative insulin sensitivity check index; SBP, systolic blood pressure; DBP, diastolic blood pressure; MDA, malondialdehyde; AMH, anti-müllerian hormone; NO, nitric oxide; GSH, glutathione; SMM, skeletal muscle mass; AIP, atherogenic index of plasma; TOS, total oxidant status.

## Data Availability

Data described in the manuscript, code book, and analytic code will be made available upon request.

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
