# Peer review of "The Effect of Prebiotics, Alone or as Part of Synbiotics, on Cardiometabolic Parameters in Women with Polycystic Ovary Syndrome: A Systematic Review and Meta-Analysis of Randomized Controlled Trials"

_biomedicines, 2025, doi:10.3390/biomedicines13010177_

Round 1
Reviewer 1 Report
Comments and Suggestions for Authors
In this systemic review and meta-analysis, the authors aimed to investigate the effects prebiotics and/or synbiotics on cardiometabolic parameters in women suffering from polycystic ovary syndrome. The manuscript is well-written and I believe that it will take attention of the readers of Biomedicines. Thus, I believe it deserves to be published. However, before publication, there are some points (as pointed below) that need to be revised:
1) In the abstract, many different abbreviations are used without description. They have to be described in the first place where they are used.
2) In the method section of abstract, the databases that were used to search should be named.
3) Line 56, please remove comma after prebiotics.
4) Lİne 58-59: Resistant starch is not considered as prebiotic according to the prebiotic definition. Please revise the sentence accordingly.
5) The background color of figures should be removed. Plain-white color should make them easier to be seen.
Author Response
Reviewer 1:
In this systemic review and meta-analysis, the authors aimed to investigate the effects prebiotics and/or synbiotics on cardiometabolic parameters in women suffering from polycystic ovary syndrome. The manuscript is well-written and I believe that it will take attention of the readers of Biomedicines. Thus, I believe it deserves to be published. However, before publication, there are some points (as pointed below) that need to be revised:
- In the abstract, many different abbreviations are used without description. They have to be described in the first place where they are used.
Response: Thank you. The full forms of the abbreviations are described in the abstract. Revisions are highlighted in yellow. Lines 15-27.
- In the method section of abstract, the databases that were used to search should be named.
Response: Thank you. The list of databases was added to the abstract. Revisions are highlighted in yellow. Lines 15-17.
- Line 56, please remove comma after prebiotics.
Response: Thank you. It has been removed.
- Line 58-59: Resistant starch is not considered as prebiotic according to the prebiotic definition. Please revise the sentence accordingly.
Response: Thank you. It has been revised. Revisions are highlighted in yellow. Lines 51-53.
- The background color of figures should be removed. Plain-white color should make them easier to be seen.
Response: The color of background figures has been removed.
Reviewer 2:
I reviewed the manuscript entitled The Effect of Prebiotics, Alone or as Part of Synbiotics, on Cardiometabolic Parameters in Women with Polycystic Ovary Syndrome: A Systematic Review and Meta-Analysis of Randomized Controlled Trials.
I agree to accept this manuscript after minor revision.
- Methods: Databases were searched for relevant RCTs until November 2023. It's already December 2024, 13 months have passed, and I think it's necessary for the author to update. If they can find it, they can add it. If they can't find it, they can extend the time until now.
Response: Thank you. Databases were searched and 136 new articles were screened. Seven articles were assessed for full-text screening. Four studies (Chudzicka-Strugała, Kubiak et al. 2024, Dou, Zhang et al. 2024, Hariri, Yari et al. 2024, Ziaei, Shahshahan et al. 2024) reported similar findings that were previously published and already included in our analysis (Chudzicka-Strugala, Kubiak et al. 2021, Ziaei, Shahshahan et al. 2022, Dou, Zhang et al. 2023, Hariri, Yari et al. 2023); therefore, they were excluded from the present study. The remaining three articles did not meet the inclusion criteria. The first study investigated the MIND diet in patients with PCOS; although the MIND diet was not part of the inclusion criteria, we considered the amount of fiber in the study; As the intervention was not a high-fiber diet and the fiber intake was lower than in the control group, the study was excluded (Kabiri, Javanbakht et al. 2024). The second study combined metformin with fenugreek seeds, a rich source of fiber, which did not meet the inclusion criteria (Mirgaloybayat, Akbari Sene et al. 2024). The third study used a combination of the DASH diet and curcumin, which also fell outside the inclusion criteria and was excluded (Zohrabi, Nadjarzadeh et al. 2024). These updates were added to the article. Revisions are highlighted in yellow. Lines 132-137.
- Subgroup analyses revealed possible increase in insulin sensitivity and reduction in LDL with prebiotics, and possible decreases in WC, TC, TG, and total testosterone with synbiotics. DASH diet improved insulin sensitivity. The author used many abbreviations in the abstract. In general, it is necessary to use abbreviations when they appear three or more times, otherwise too many abbreviations will confuse readers. Please revise the abstract and the entire text according to this principle.
Response: Thank you. The full forms of the abbreviations are described in the abstract. Revisions are highlighted in yellow. Lines 15-27.
- Keywords, except for the first keyword, which needs to be capitalized, the first letters of all other keywords should be lowercase.
Response: Thank you for your clarification. The keywords have been updated accordingly, with only the first keyword capitalized and the first letters of all other keywords in lowercase. Revisions are highlighted in yellow. Line 28.
- Table 1. First author, year (reference) should change to Reference, The following content should be written as, for example, [18]. This modification meets the requirements of the journal.
Response: Thank you for the guidance. The requested changes to Table 1 have been made, with "First author, year (reference)" updated to "Reference. Pages 11-21.
- 4.1. Primary Outcomes. 13 RCTs that included 801 participants… Suggest modifying it to thirteen RCTs that included 801 participants… This modification makes it clearer. Please review and revise the entire text.
Response: Thank you for the suggestion. The revision has been completed, and the entire text has been updated accordingly. Revisions are highlighted in yellow. Lines 222-246 and lines 248-293.
- P<0.001; When it comes to statistics, P should be italicized, please review and revise the entire text.
Response: Thank you. The formatting has been updated, and all instances of "P" have been italicized as per statistical conventions. The entire text has been reviewed and revised accordingly, including figure 1 and figure 2.
- Inflammatory and oxidative stress markers [High-sensitivity c-reactive protein (hs-CRP), Total antioxidant capacity (TAC), Malondialdehyde (MDA), and Nitric oxide (NO)]. The first letters of words in parentheses do not need to be capitalized.
Response: Thank you for the suggestion. The first letters of the words in parentheses have been changed to lowercase, and the text has been revised accordingly. Revisions are highlighted in yellow. Lines 238-239.
- We additionally performed a sensitivity analysis, excluding articles of very low and low quality based on the GRADE evaluation. The results revealed no significant changes in any of the parameters A period is missing at the end of the sentence, please add it.
Response: Thank you. It had been added.
- There are multiple uses of 'we' and 'our' in the article. It is recommended to revise them. If necessary, they can be used, but if not, they should not be used. This will increase the scientific and objective nature of the article after revision.
Response: Thank you for the suggestion. The text has been revised to minimize the use of "we" and "our," ensuring a more scientific and objective tone throughout the article. Revisions are highlighted in yellow. Lines 311-380.
- This extensive initial systematic review and meta-analysis of RCTs indicates that prebiotics, either alone or as part of synbiotics, could enhance cardiometabolic health in overweight or obese adult women with PCOS by improving factors such as BMI, WC, WHR, body weight, fat mass, TG, and DBP. Furthermore, the study suggests potential improvements in glycemic, lipid, hormonal, inflammatory, and antioxidant markers, including FPG, fasting insulin, LDL, TC, TG, hs-CRP, insulin sensitivity, NO and total testosterone. However, despite providing valuable information, the study emphasizes the need for larger, well-designed RCTs with varying types and doses of prebiotics, targeting different PCOS phenotypes, and conducted over a longer duration to further strengthen the evidence and inform clinical guidelines.
Response: Thank you for your comment.
- Several prior systematic reviews and meta-analyses have primarily focused on investigating the influence of probiotics or synbiotics on polycystic ovary syndrome (PCOS), yielding mixed results. Notably, these studies have lacked a thorough examination of the specific effects of prebiotics within PCOS management. In contrast, this study represents a significant advancement, being the inaugural systematic review and meta-analysis to examine the impact of prebiotics and synbiotics, considered as sources of prebiotics, on multiple clinical outcomes in PCOS patients. They meticulously included all randomized controlled trials (RCTs) exploring the effects of prebiotics, synbiotics, and high-fiber interventions potentially acting as prebiotics. Additionally, they utilized the GRADE tool to assess the certainty of evidence, enhancing the clarity and validity of their findings, thereby addressing a critical gap in the existing research.
Response: Thank you for your comment.
- The conclusion is consistent with the evidence and arguments provided. All the main questions raised by the author have been resolved.
Response: Thank you for your comment.
- The biggest problem with this study is the issue of not being able to retrieve the latest literature. Readers all want to read the latest research results, but the author only has until November of the previous year.
Response: Thank you. Databases were searched and new articles were screened. Seven articles were assessed for full-text screening. Four studies (Chudzicka-Strugała, Kubiak et al. 2024, Dou, Zhang et al. 2024, Hariri, Yari et al. 2024, Ziaei, Shahshahan et al. 2024) reported similar findings that were previously published and already included in our analysis (Chudzicka-Strugala, Kubiak et al. 2021, Ziaei, Shahshahan et al. 2022, Dou, Zhang et al. 2023, Hariri, Yari et al. 2023); therefore, they were excluded from the present study. The remaining three articles did not meet the inclusion criteria. The first study investigated the MIND diet in patients with PCOS; although the MIND diet was not part of the inclusion criteria, we considered the amount of fiber in the study; As the intervention was not a high-fiber diet and the fiber intake was lower than in the control group, the study was excluded (Kabiri, Javanbakht et al. 2024). The second study combined metformin with fenugreek seeds, a rich source of fiber, which did not meet the inclusion criteria (Mirgaloybayat, Akbari Sene et al. 2024). The third study used a combination of the DASH diet and curcumin, which also fell outside the inclusion criteria and was excluded (Zohrabi, Nadjarzadeh et al. 2024). These updates were added to the article. Revisions are highlighted in yellow. Lines 132-137.
- I have read all the references and found some issues. Ref 2, missing start and end page numbers; Some journal names are complete, while others are abbreviated. It is recommended to use abbreviations uniformly. The website for standard abbreviations is https://cassi.cas.org/search.jsp Please search and modify. Ref 50, missing DOI number.
Response: Thank you. References have been revised. Reference no. 50 is an abstract that did not have a Doi. This is one of the two studies initially existed as abstracts (Karamali, Samimi et al. 2014, Hosseinzadeh-Attar, Karimi et al. 2020), later published in full texts (Asemi and Esmaillzadeh 2015, Karimi, Moini et al. 2018).
References :
Asemi, Z. and A. Esmaillzadeh (2015). "DASH Diet, Insulin Resistance, and Serum hs-CRP in Polycystic Ovary Syndrome: A Randomized Controlled Clinical Trial." Hormone and Metabolic Research 47(3): 232-238.
Chudzicka-Strugała, I., A. Kubiak, B. Banaszewska, E. Wysocka, B. Zwozdziak, M. Siakowska, L. Pawelczyk and A. J. Duleba (2024). "Six-month randomized, placebo controlled trial of synbiotic supplementation in women with polycystic ovary syndrome undergoing lifestyle modifications." Arch Gynecol Obstet.
Chudzicka-Strugala, I., A. Kubiak, B. Banaszewska, B. Zwozdziak, M. Siakowska, L. Pawelczyk and A. J. Duleba (2021). "Effects of Synbiotic Supplementation and Lifestyle Modifications on Women With Polycystic Ovary Syndrome." Journal of Clinical Endocrinology & Metabolism 106(9): 2566-2573.
Dou, P., T. T. Zhang, Y. Xu, Q. Xue, J. Shang and X. L. Yang (2023). "[Effects of three medical nutrition therapies for weight loss on metabolic parameters and androgen level in overweight/obese patients with polycystic ovary syndrome]." Zhonghua Yi Xue Za Zhi 103(14): 1035-1041.
Dou, P., T. T. Zhang, Y. Xu, Q. Xue, Y. Zhang, J. Shang and X. L. Yang (2024). "A Randomized Trial of the Efficacy of Three Weight Loss Diet Interventions in Overweight/Obese with Polycystic Ovary Syndrome." Endocr Metab Immune Disord Drug Targets 24(14): 1686-1697.
Hariri, Z., Z. Yari, S. Hoseini, K. Abhari and G. Sohrab (2024). "Synbiotic as an ameliorating factor in the health-related quality of life in women with polycystic ovary syndrome. A randomized, triple-blind, placebo-controlled trial." BMC Womens Health 24(1): 19.
Hariri, Z., Z. Yari, S. Hoseini, A. Mehrnami, K. Abhari and G. Sohrab (2023). "Effects of Synbiotic-Containing Bacillus coagulans (GBI-30) on the Cardiovascular Status of Patients With Polycystic Ovary Syndrome: A Triple-blinded, Randomized, Placebo-controlled Study." Clin Ther.
Hosseinzadeh-Attar, M. J., E. Karimi, E. Alipoor and A. Moini (2020). "Effect of a Synbiotic Supplement on Cardiovascular Risk Factors in Overweight or Obese Patients with Polycystic Ovary Syndrome." Journal of Clinical Gastroenterology 54: S26-S26.
Kabiri, S. S., Z. Javanbakht, M. Zangeneh, J. Moludi, A. Saber, Y. Salimi, A. Tandorost and M. Jamalpour (2024). "The effects of MIND diet on depression, anxiety, quality of life and metabolic and hormonal status in obese or overweight women with polycystic ovary syndrome: a randomised clinical trial." Br J Nutr: 1-14.
Karamali, M., M. Samimi, F. Bahmani, F. Foroozanfard and A. Esmaillzadeh (2014). "The effects of DASH diet on lipid profiles and biomarkers of oxidative stress in overweight and obese women with polycystic ovary syndrome: a randomised clinical trial." Human reproduction (Oxford, England) 29: i315‐i316.
Karimi, E., A. Moini, M. Yaseri, N. Shirzad, M. Sepidarkish, M. Hossein-Boroujerdi and M. J. Hosseinzadeh-Attar (2018). "Erratum: Effects of synbiotic supplementation on metabolic parameters and apelin in women with polycystic ovary syndrome: A randomised double-blind placebo-controlled trial (British Journal of Nutrition DOI: 10.1017/S0007114517003920)." British Journal of Nutrition 124(4): 479.
Mirgaloybayat, S., A. Akbari Sene, F. Jayervand, M. Vazirian, A. Mohazzab and M. Kazerooni (2024). "Comparison of the Effect of Fenugreek and Metformin on Clinical and Metabolic Status of Cases with Polycystic Ovary Syndrome: A Randomized Trial." J Reprod Infertil 25(2): 120-132.
Ziaei, R., Z. Shahshahan, H. Ghasemi-Tehrani, Z. Heidari and R. Ghiasvand (2022). "Effects of inulin-type fructans with different degrees of polymerization on inflammation, oxidative stress and endothelial dysfunction in women with polycystic ovary syndrome: A randomized, double-blind, placebo-controlled trial." Clinical Endocrinology 97(3): 319-330.
Ziaei, R., Z. Shahshahan, H. Ghasemi-Tehrani, Z. Heidari, M. S. Nehls and R. Ghiasvand (2024). "Inulin-type fructans with different degrees of polymerization improve insulin resistance, metabolic parameters, and hormonal status in overweight and obese women with polycystic ovary syndrome: A randomized double-blind, placebo-controlled clinical trial." Food Sci Nutr 12(3): 2016-2028.
Zohrabi, T., A. Nadjarzadeh, S. Jambarsang, M. H. Sheikhha, A. Aflatoonian and H. Mozaffari-Khosravi (2024). "Effect of dietary approaches to stop hypertension and curcumin co-administration on glycemic parameters in polycystic ovary syndrome: An RCT." Int J Reprod Biomed 22(9): 689-700.
Reviewer 2 Report
Comments and Suggestions for Authors
I reviewed the manuscript entitled The Effect of Prebiotics, Alone or as Part of Synbiotics, on Cardiometabolic Parameters in Women with Polycystic Ovary Syndrome: A Systematic Review and Meta-Analysis of Randomized Controlled Trials.
I agree to accept this manuscript after minor revision.
1) Methods: Databases were searched for relevant RCTs until November 2023. It's already December 2024, 13 months have passed, and I think it's necessary for the author to update. If they can find it, they can add it. If they can't find it, they can extend the time until now.
2) Subgroup analyses revealed possible increase in insulin sensitivity and reduction in LDL with prebiotics, and possible decreases in WC, TC, TG, and total testosterone with synbiotics. DASH diet improved insulin sensitivity. The author used many abbreviations in the abstract. In general, it is necessary to use abbreviations when they appear three or more times, otherwise too many abbreviations will confuse readers. Please revise the abstract and the entire text according to this principle.
3) Keywords, except for the first keyword, which needs to be capitalized, the first letters of all other keywords should be lowercase.
4) Table 1. First author, year (reference) should change to Reference, The following content should be written as, for example, [18]. This modification meets the requirements of the journal.
5) 3.4.1. Primary Outcomes. 13 RCTs that included 801 participants… Suggest modifying it to thirteen RCTs that included 801 participants… This modification makes it clearer. Please review and revise the entire text.
6) P<0.001; When it comes to statistics, P should be italicized, please review and revise the entire text.
7) Inflammatory and oxidative stress markers [High-sensitivity c-reactive protein (hs-CRP), Total antioxidant capacity (TAC), Malondialdehyde (MDA), and Nitric oxide (NO)]. The first letters of words in parentheses do not need to be capitalized.
8) We additionally performed a sensitivity analysis, excluding articles of very low and low quality based on the GRADE evaluation. The results revealed no significant changes in any of the parameters A period is missing at the end of the sentence, please add it.
9) There are multiple uses of 'we' and 'our' in the article. It is recommended to revise them. If necessary, they can be used, but if not, they should not be used. This will increase the scientific and objective nature of the article after revision.
10) This extensive initial systematic review and meta-analysis of RCTs indicates that prebiotics, either alone or as part of synbiotics, could enhance cardiometabolic health in overweight or obese adult women with PCOS by improving factors such as BMI, WC, WHR, body weight, fat mass, TG, and DBP. Furthermore, the study suggests potential improvements in glycemic, lipid, hormonal, inflammatory, and antioxidant markers, including FPG, fasting insulin, LDL, TC, TG, hs-CRP, insulin sensitivity, NO and total testosterone. However, despite providing valuable information, the study emphasizes the need for larger, well-designed RCTs with varying types and doses of prebiotics, targeting different PCOS phenotypes, and conducted over a longer duration to further strengthen the evidence and inform clinical guidelines.
11) Several prior systematic reviews and meta-analyses have primarily focused on investigating the influence of probiotics or synbiotics on polycystic ovary syndrome (PCOS), yielding mixed results. Notably, these studies have lacked a thorough examination of the specific effects of prebiotics within PCOS management. In contrast, this study represents a significant advancement, being the inaugural systematic review and meta-analysis to examine the impact of prebiotics and synbiotics, considered as sources of prebiotics, on multiple clinical outcomes in PCOS patients. They meticulously included all randomized controlled trials (RCTs) exploring the effects of prebiotics, synbiotics, and high-fiber interventions potentially acting as prebiotics. Additionally, they utilized the GRADE tool to assess the certainty of evidence, enhancing the clarity and validity of their findings, thereby addressing a critical gap in the existing research.
12) The conclusion is consistent with the evidence and arguments provided. All the main questions raised by the author have been resolved.
13) The biggest problem with this study is the issue of not being able to retrieve the latest literature. Readers all want to read the latest research results, but the author only has until November of the previous year.
14) I have read all the references and found some issues. Ref 2, missing start and end page numbers; Some journal names are complete, while others are abbreviated. It is recommended to use abbreviations uniformly. The website for standard abbreviations is https://cassi.cas.org/search.jsp Please search and modify. Ref 50, missing DOI number.
Author Response
I reviewed the manuscript entitled The Effect of Prebiotics, Alone or as Part of Synbiotics, on Cardiometabolic Parameters in Women with Polycystic Ovary Syndrome: A Systematic Review and Meta-Analysis of Randomized Controlled Trials.
I agree to accept this manuscript after minor revision.
- Methods: Databases were searched for relevant RCTs until November 2023. It's already December 2024, 13 months have passed, and I think it's necessary for the author to update. If they can find it, they can add it. If they can't find it, they can extend the time until now.
Response: Thank you. Databases were searched and 136 new articles were screened. Seven articles were assessed for full-text screening. Four studies (Chudzicka-Strugała, Kubiak et al. 2024, Dou, Zhang et al. 2024, Hariri, Yari et al. 2024, Ziaei, Shahshahan et al. 2024) reported similar findings that were previously published and already included in our analysis (Chudzicka-Strugala, Kubiak et al. 2021, Ziaei, Shahshahan et al. 2022, Dou, Zhang et al. 2023, Hariri, Yari et al. 2023); therefore, they were excluded from the present study. The remaining three articles did not meet the inclusion criteria. The first study investigated the MIND diet in patients with PCOS; although the MIND diet was not part of the inclusion criteria, we considered the amount of fiber in the study; As the intervention was not a high-fiber diet and the fiber intake was lower than in the control group, the study was excluded (Kabiri, Javanbakht et al. 2024). The second study combined metformin with fenugreek seeds, a rich source of fiber, which did not meet the inclusion criteria (Mirgaloybayat, Akbari Sene et al. 2024). The third study used a combination of the DASH diet and curcumin, which also fell outside the inclusion criteria and was excluded (Zohrabi, Nadjarzadeh et al. 2024). These updates were added to the article. Revisions are highlighted in yellow. Lines 132-137.
- Subgroup analyses revealed possible increase in insulin sensitivity and reduction in LDL with prebiotics, and possible decreases in WC, TC, TG, and total testosterone with synbiotics. DASH diet improved insulin sensitivity. The author used many abbreviations in the abstract. In general, it is necessary to use abbreviations when they appear three or more times, otherwise too many abbreviations will confuse readers. Please revise the abstract and the entire text according to this principle.
Response: Thank you. The full forms of the abbreviations are described in the abstract. Revisions are highlighted in yellow. Lines 15-27.
- Keywords, except for the first keyword, which needs to be capitalized, the first letters of all other keywords should be lowercase.
Response: Thank you for your clarification. The keywords have been updated accordingly, with only the first keyword capitalized and the first letters of all other keywords in lowercase. Revisions are highlighted in yellow. Line 28.
- Table 1. First author, year (reference) should change to Reference, The following content should be written as, for example, [18]. This modification meets the requirements of the journal.
Response: Thank you for the guidance. The requested changes to Table 1 have been made, with "First author, year (reference)" updated to "Reference. Pages 11-21.
- 4.1. Primary Outcomes. 13 RCTs that included 801 participants… Suggest modifying it to thirteen RCTs that included 801 participants… This modification makes it clearer. Please review and revise the entire text.
Response: Thank you for the suggestion. The revision has been completed, and the entire text has been updated accordingly. Revisions are highlighted in yellow. Lines 222-246 and lines 248-293.
- P<0.001; When it comes to statistics, P should be italicized, please review and revise the entire text.
Response: Thank you. The formatting has been updated, and all instances of "P" have been italicized as per statistical conventions. The entire text has been reviewed and revised accordingly, including figure 1 and figure 2.
- Inflammatory and oxidative stress markers [High-sensitivity c-reactive protein (hs-CRP), Total antioxidant capacity (TAC), Malondialdehyde (MDA), and Nitric oxide (NO)]. The first letters of words in parentheses do not need to be capitalized.
Response: Thank you for the suggestion. The first letters of the words in parentheses have been changed to lowercase, and the text has been revised accordingly. Revisions are highlighted in yellow. Lines 238-239.
- We additionally performed a sensitivity analysis, excluding articles of very low and low quality based on the GRADE evaluation. The results revealed no significant changes in any of the parameters A period is missing at the end of the sentence, please add it.
Response: Thank you. It had been added.
- There are multiple uses of 'we' and 'our' in the article. It is recommended to revise them. If necessary, they can be used, but if not, they should not be used. This will increase the scientific and objective nature of the article after revision.
Response: Thank you for the suggestion. The text has been revised to minimize the use of "we" and "our," ensuring a more scientific and objective tone throughout the article. Revisions are highlighted in yellow. Lines 311-380.
- This extensive initial systematic review and meta-analysis of RCTs indicates that prebiotics, either alone or as part of synbiotics, could enhance cardiometabolic health in overweight or obese adult women with PCOS by improving factors such as BMI, WC, WHR, body weight, fat mass, TG, and DBP. Furthermore, the study suggests potential improvements in glycemic, lipid, hormonal, inflammatory, and antioxidant markers, including FPG, fasting insulin, LDL, TC, TG, hs-CRP, insulin sensitivity, NO and total testosterone. However, despite providing valuable information, the study emphasizes the need for larger, well-designed RCTs with varying types and doses of prebiotics, targeting different PCOS phenotypes, and conducted over a longer duration to further strengthen the evidence and inform clinical guidelines.
Response: Thank you for your comment.
- Several prior systematic reviews and meta-analyses have primarily focused on investigating the influence of probiotics or synbiotics on polycystic ovary syndrome (PCOS), yielding mixed results. Notably, these studies have lacked a thorough examination of the specific effects of prebiotics within PCOS management. In contrast, this study represents a significant advancement, being the inaugural systematic review and meta-analysis to examine the impact of prebiotics and synbiotics, considered as sources of prebiotics, on multiple clinical outcomes in PCOS patients. They meticulously included all randomized controlled trials (RCTs) exploring the effects of prebiotics, synbiotics, and high-fiber interventions potentially acting as prebiotics. Additionally, they utilized the GRADE tool to assess the certainty of evidence, enhancing the clarity and validity of their findings, thereby addressing a critical gap in the existing research.
Response: Thank you for your comment.
- The conclusion is consistent with the evidence and arguments provided. All the main questions raised by the author have been resolved.
Response: Thank you for your comment.
- The biggest problem with this study is the issue of not being able to retrieve the latest literature. Readers all want to read the latest research results, but the author only has until November of the previous year.
Response: Thank you. Databases were searched and new articles were screened. Seven articles were assessed for full-text screening. Four studies (Chudzicka-Strugała, Kubiak et al. 2024, Dou, Zhang et al. 2024, Hariri, Yari et al. 2024, Ziaei, Shahshahan et al. 2024) reported similar findings that were previously published and already included in our analysis (Chudzicka-Strugala, Kubiak et al. 2021, Ziaei, Shahshahan et al. 2022, Dou, Zhang et al. 2023, Hariri, Yari et al. 2023); therefore, they were excluded from the present study. The remaining three articles did not meet the inclusion criteria. The first study investigated the MIND diet in patients with PCOS; although the MIND diet was not part of the inclusion criteria, we considered the amount of fiber in the study; As the intervention was not a high-fiber diet and the fiber intake was lower than in the control group, the study was excluded (Kabiri, Javanbakht et al. 2024). The second study combined metformin with fenugreek seeds, a rich source of fiber, which did not meet the inclusion criteria (Mirgaloybayat, Akbari Sene et al. 2024). The third study used a combination of the DASH diet and curcumin, which also fell outside the inclusion criteria and was excluded (Zohrabi, Nadjarzadeh et al. 2024). These updates were added to the article. Revisions are highlighted in yellow. Lines 132-137.
- I have read all the references and found some issues. Ref 2, missing start and end page numbers; Some journal names are complete, while others are abbreviated. It is recommended to use abbreviations uniformly. The website for standard abbreviations is https://cassi.cas.org/search.jsp Please search and modify. Ref 50, missing DOI number.
Response: Thank you. References have been revised. Reference no. 50 is an abstract that did not have a Doi. This is one of the two studies initially existed as abstracts (Karamali, Samimi et al. 2014, Hosseinzadeh-Attar, Karimi et al. 2020), later published in full texts (Asemi and Esmaillzadeh 2015, Karimi, Moini et al. 2018).
References :
Asemi, Z. and A. Esmaillzadeh (2015). "DASH Diet, Insulin Resistance, and Serum hs-CRP in Polycystic Ovary Syndrome: A Randomized Controlled Clinical Trial." Hormone and Metabolic Research 47(3): 232-238.
Chudzicka-Strugała, I., A. Kubiak, B. Banaszewska, E. Wysocka, B. Zwozdziak, M. Siakowska, L. Pawelczyk and A. J. Duleba (2024). "Six-month randomized, placebo controlled trial of synbiotic supplementation in women with polycystic ovary syndrome undergoing lifestyle modifications." Arch Gynecol Obstet.
Chudzicka-Strugala, I., A. Kubiak, B. Banaszewska, B. Zwozdziak, M. Siakowska, L. Pawelczyk and A. J. Duleba (2021). "Effects of Synbiotic Supplementation and Lifestyle Modifications on Women With Polycystic Ovary Syndrome." Journal of Clinical Endocrinology & Metabolism 106(9): 2566-2573.
Dou, P., T. T. Zhang, Y. Xu, Q. Xue, J. Shang and X. L. Yang (2023). "[Effects of three medical nutrition therapies for weight loss on metabolic parameters and androgen level in overweight/obese patients with polycystic ovary syndrome]." Zhonghua Yi Xue Za Zhi 103(14): 1035-1041.
Dou, P., T. T. Zhang, Y. Xu, Q. Xue, Y. Zhang, J. Shang and X. L. Yang (2024). "A Randomized Trial of the Efficacy of Three Weight Loss Diet Interventions in Overweight/Obese with Polycystic Ovary Syndrome." Endocr Metab Immune Disord Drug Targets 24(14): 1686-1697.
Hariri, Z., Z. Yari, S. Hoseini, K. Abhari and G. Sohrab (2024). "Synbiotic as an ameliorating factor in the health-related quality of life in women with polycystic ovary syndrome. A randomized, triple-blind, placebo-controlled trial." BMC Womens Health 24(1): 19.
Hariri, Z., Z. Yari, S. Hoseini, A. Mehrnami, K. Abhari and G. Sohrab (2023). "Effects of Synbiotic-Containing Bacillus coagulans (GBI-30) on the Cardiovascular Status of Patients With Polycystic Ovary Syndrome: A Triple-blinded, Randomized, Placebo-controlled Study." Clin Ther.
Hosseinzadeh-Attar, M. J., E. Karimi, E. Alipoor and A. Moini (2020). "Effect of a Synbiotic Supplement on Cardiovascular Risk Factors in Overweight or Obese Patients with Polycystic Ovary Syndrome." Journal of Clinical Gastroenterology 54: S26-S26.
Kabiri, S. S., Z. Javanbakht, M. Zangeneh, J. Moludi, A. Saber, Y. Salimi, A. Tandorost and M. Jamalpour (2024). "The effects of MIND diet on depression, anxiety, quality of life and metabolic and hormonal status in obese or overweight women with polycystic ovary syndrome: a randomised clinical trial." Br J Nutr: 1-14.
Karamali, M., M. Samimi, F. Bahmani, F. Foroozanfard and A. Esmaillzadeh (2014). "The effects of DASH diet on lipid profiles and biomarkers of oxidative stress in overweight and obese women with polycystic ovary syndrome: a randomised clinical trial." Human reproduction (Oxford, England) 29: i315‐i316.
Karimi, E., A. Moini, M. Yaseri, N. Shirzad, M. Sepidarkish, M. Hossein-Boroujerdi and M. J. Hosseinzadeh-Attar (2018). "Erratum: Effects of synbiotic supplementation on metabolic parameters and apelin in women with polycystic ovary syndrome: A randomised double-blind placebo-controlled trial (British Journal of Nutrition DOI: 10.1017/S0007114517003920)." British Journal of Nutrition 124(4): 479.
Mirgaloybayat, S., A. Akbari Sene, F. Jayervand, M. Vazirian, A. Mohazzab and M. Kazerooni (2024). "Comparison of the Effect of Fenugreek and Metformin on Clinical and Metabolic Status of Cases with Polycystic Ovary Syndrome: A Randomized Trial." J Reprod Infertil 25(2): 120-132.
Ziaei, R., Z. Shahshahan, H. Ghasemi-Tehrani, Z. Heidari and R. Ghiasvand (2022). "Effects of inulin-type fructans with different degrees of polymerization on inflammation, oxidative stress and endothelial dysfunction in women with polycystic ovary syndrome: A randomized, double-blind, placebo-controlled trial." Clinical Endocrinology 97(3): 319-330.
Ziaei, R., Z. Shahshahan, H. Ghasemi-Tehrani, Z. Heidari, M. S. Nehls and R. Ghiasvand (2024). "Inulin-type fructans with different degrees of polymerization improve insulin resistance, metabolic parameters, and hormonal status in overweight and obese women with polycystic ovary syndrome: A randomized double-blind, placebo-controlled clinical trial." Food Sci Nutr 12(3): 2016-2028.
Zohrabi, T., A. Nadjarzadeh, S. Jambarsang, M. H. Sheikhha, A. Aflatoonian and H. Mozaffari-Khosravi (2024). "Effect of dietary approaches to stop hypertension and curcumin co-administration on glycemic parameters in polycystic ovary syndrome: An RCT." Int J Reprod Biomed 22(9): 689-700.